# UMamba-ProSSL: Self-Supervised Large-Scale Pretraining with Multi-Task UMamba Advances Prostate Cancer Detection in Biparametric MRI

**Syed Farhan Abbas**[1]                                                     SYED.F.ABBAS@NTNU.NO
**Michael S. Larsen**[1,2]                                             MICHAEL.S.LARSEN@NTNU.NO
**Arild Strømsvåg**[2]                                                  ARILDSTROM@OUTLOOK.COM
**Tone F. Bathen**[1]                                                   TONE.F.BATHEN@NTNU.NO
**Frank Lindseth**[2]                                                              FRANKL@NTNU.NO
**Gabriel Kiss**[2]                                                        GABRIEL.KISS@NTNU.NO
**Mattijs Elschot**[1,3]                                               MATTIJS.ELSCHOT@NTNU.NO

[1] *Department of Circulation and Medical Imaging, Norwegian University of Science and Technology*

[2] *Department of Computer Science, Norwegian University of Science and Technology*

[3] *Central Staff, St. Olavs Hospital, Trondheim University Hospital*

**Editors:** Accepted for publication at MIDL 2026

## Abstract

Accurate prostate cancer (PCa) diagnosis is crucial, as it remains one of the leading cause of mortality among men. Although prostate magnetic resonance imaging (MRI) has improved the diagnostic workflow, radiologists still face challenges due to inter-observer variability and limited specificity, leading to both over- and under-diagnosis. Deep learning methods have the potential to support radiologists, but their performance typically depends on large, high-quality labeled datasets that are often scarce and expensive to curate. In contrast, large volumes of unlabeled prostate MRI scans are routinely generated in clinical practice, making self-supervised learning (SSL) a compelling approach to exploit this abundant, untapped resource. However, SSL performance depends strongly on backbone architectures and effective pretext tasks. Moreover, the lack of large-scale standardized benchmarking further limits progress. In this study, we employ a state-of-the-art UMamba for prostate cancer detection and investigate several SSL strategies using a large in-house unlabeled prostate MRI dataset (N=2,431). Among the different pretraining methods, UMamba pretrained with masked autoencoders (MAE) achieved the best downstream performance, with an aggregated mean score of **0.780** (AUROC: 0.905, AP: 0.655) on the large-scale PI-CAI hidden testing set (N=1,000). This performance ranked **first** on the PI-CAI benchmark leaderboard at the time of evaluation. To further evaluate generalizability, we conducted an evaluation on the out-of-distribution Prostate158 (N=158) dataset, where MAE-pretrained UMamba achieved the best generalization performance, indicating robustness across different clinical centers and imaging protocols. These findings highlighting the strong potential of SSL, particularly MAE combined with UMamba for improving PCa detection accuracy and potentially reducing unnecessary biopsies. The code is available at https://github.com/farhancv09/UMamba-ProSSL.

**Keywords:** Self-Supervised Learning, Prostate Cancer (PCa), UMamba, Masked Autoencoders, Magnetic Resonance Imaging

## 1. Introduction

Prostate cancer is the second most common cancer in men, fourth overall, and the eighth leading cause of cancer-related deaths worldwide (Bray et al., 2024). While histologically confirmed biopsies remain the gold standard for diagnosis and grading tumor aggressiveness, they are invasive, prone to overdiagnosis of clinically insignificant cancers, underdiagnosis of clinically significant cancers, and carry risks of infection and sepsis (Loeb et al., 2013; Borghesi et al., 2017). Prostate-specific antigen (PSA) testing and digital rectal examination (DRE) are commonly used for screening, but both have limited specificity and can lead to unnecessary follow-up procedures (Grossman et al., 2018; Jones et al., 2018). To improve the detection of clinically significant prostate cancer (csPCa), multiparametric magnetic resonance imaging (mpMRI), which includes $T_2$-weighted ($T_2$w) imaging, diffusion-weighted imaging (DWI) with high b-value (HBV) and apparent diffusion coefficient (ADC) maps, and dynamic contrast-enhanced (DCE) imaging has become the standard pre-biopsy method (EAU, 2019; Dasgupta et al., 2019). To address concerns about contrast agent use and scan time, bi-parametric MRI (bpMRI), which excludes DCE, has gained clinical traction while maintaining comparable diagnostic performance to mpMRI (Tamada et al., 2021; Twilt et al., 2025). Its interpretation is guided by the prostate imaging-reporting and data system version 2.1 (PI-RADS v2.1) (Park et al., 2021). However, diagnostic accuracy still depends on radiologist expertise and suffers from inter-reader variability (Wei et al., 2021).

AI-driven computer-aided diagnosis (CAD) systems reduce inter-reader variability in prostate MRI through automated solutions. To support the development, fair evaluation, and comparison of such algorithms, grand challenges[1] provide unbiased and standardized benchmarking using hidden test sets (van Leeuwen et al., 2021). Earlier efforts for PCa detection, like ProstateX (Armato III et al., 2018), were limited by small, single-center datasets. To overcome these limitations, the multi-center Prostate Imaging–Cancer Artificial Intelligence (PI-CAI) challenge (Saha et al., 2024) included 10,207 prostate MRI scans from 9,129 patients across four centers, showing that deep learning ensembles outperformed radiologists. These results highlight the potential of scalable deep learning approaches in prostate MRI analysis. However, assembling such large, expertly annotated datasets is resource-intensive, and most clinical centers have far more unlabeled prostate MRI available, highlighting the need for scalable learning strategies.

Self-supervised learning (SSL) has gained traction in large data domains, enabling pretraining using a pretext task to learn robust feature representations from data. Pretext tasks are self-supervised objectives, such as masked restoration (He et al., 2022) or contrastive learning (Chen et al., 2020) that allow models to learn useful features without labels. While SSL has advanced fields like NLP (Achiam et al., 2023) and natural imaging (Siméoni et al., 2025), its uptake in 3D medical imaging remains limited. The field still relies heavily on training from scratch or costly supervised pretraining (Isensee et al., 2021), indicating a need for unlabeled pretraining. However, SSL adoption has lagged because prior 3D medical SSL studies often rely on small training datasets, suboptimal or outdated backbone choices compared to strong CNNs like nnU-Net (Isensee et al., 2021), and evaluations that lack robust baselines and diverse testing, ultimately hindering gener-

---

1. https://grand-challenge.org/

alizability. To address these gaps in 3D medical imaging, (Wald et al., 2025) proposed a comprehensive framework that overcomes the afore-mentioned limitations, demonstrating that masked autoencoders (MAE) combined with a strong backbone architecture outperform other self-supervised pretext tasks (Zhou et al., 2021; Wang et al., 2023; He et al., 2022; Wu et al., 2024; Tang et al., 2022; Chen et al., 2023). Given the domain-specific nature of prostate MRI, different anatomy, imaging characteristics, and lesion distribution, it remains unclear whether these SSL findings translate to csPCa detection. This motivated us to investigate multiple pretext tasks in combination with strong architectural backbones and to benchmark them externally on the PI-CAI grand challenge.

SSL has been explored to some extent in the prostate cancer (PCa) domain. Early work by (Bolous et al., 2021) employed a restoration-based patch-swapping task, but performance was limited by the small training dataset. (Li et al., 2025) introduced a transformer-based contrastive learning framework inspired by SimCLR (Chen et al., 2020) with a multi-task objective; however, the method lacks comprehensive benchmarking across diverse datasets. Large-scale SSL efforts, such as (de Almeida et al., 2025), demonstrated improvements in PCa classification, but these approaches do not address lesion localization, which is crucial for MRI-guided biopsies. A transformer-based prostate-specific foundation model proposed by (Lee et al., 2025) incorporated label-assisted pre-training and evaluated both internal and external cohorts, yet it still relies on supervised annotations. (Yuan et al., 2025) proposed a restoration-based pretext task inspired by Zhou et al. (Zhou et al., 2021), showing promising performance on the PI-CAI benchmark (Saha et al., 2024). In line with the need for stronger backbone architectures in the PCa detection domain, we proposed a UMamba–based multi-task learning model (UMamba-MTL) (Ma et al., 2024; Larsen et al., 2025) and benchmarked it against conventional CNN backbones such as nnU-Net (Isensee et al., 2021), as well as hybrid CNN-Transformer approaches like Swin-UNETR (Hatamizadeh et al., 2021).

Motivated by the need for unlabeled pretraining, we hypothesize that pairing large-scale unlabeled bpMRI with the top-performing SSL pretext tasks, identified by (Wald et al., 2025), can enable substantially improved performance for PCa detection. To test this hypothesis, we pretrain the UMamba backbone (Larsen et al., 2025) using these SSL objectives on a large in-house unlabeled dataset, and evaluate the resulting models on the PI-CAI benchmark alongside the out-of-distribution (OOD) Prostate158 dataset (Adams et al., 2022).

Our main contributions in support of this hypothesis are as follows:

- We leverage a large-scale unlabeled bpMRI dataset **(N=2,431)** for SSL pretraining, combined with publicly available datasets, and evaluate it on a large hidden cohort, making this, to our knowledge, the first large-scale SSL (**UMamba**-based **P**rostate **S**elf-**S**upervised **L**earning; **UMamba-ProSSL**) framework for csPCa lesion detection.

- We systematically explore multiple state-of-the-art pretext tasks under identical training conditions to identify the most effective SSL strategy and find that **MAE** yields the most effective transfer-learning gains.

- Our proposed architecture (UMamba-ProSSL) integrates self-supervised MAE pretraining with multi-task fine-tuning and achieves **first place** in the international PI-

CAI grand challenge benchmark and demonstrates robust performance on the external out-of-distribution P158 dataset.

## 2. Methodology

The overall framework, illustrated in Figure 1, consists of two stages: self-supervised pre-training on a large-scale unlabeled prostate bpMRI dataset, followed by supervised fine-tuning on publicly available labeled data. Both stages utilize our state-of-the-art (SOTA) architecture for prostate cancer detection, as proposed in (Larsen et al., 2025). Specifically, we employ three different pretext tasks: Volume Fusion (Wang et al., 2023), Model Genesis (Zhou et al., 2021), and Masked Autoencoders (MAE) (He et al., 2022), using the UMamba architecture (Ma et al., 2024). The learned representations are then fine-tuned for the prostate cancer detection task using our multi-task, prostate zone-aware model, and compared against Swin-UNETR, the vanilla UMamba, and Spark3D, a recent SOTA method in medical self-supervised learning.

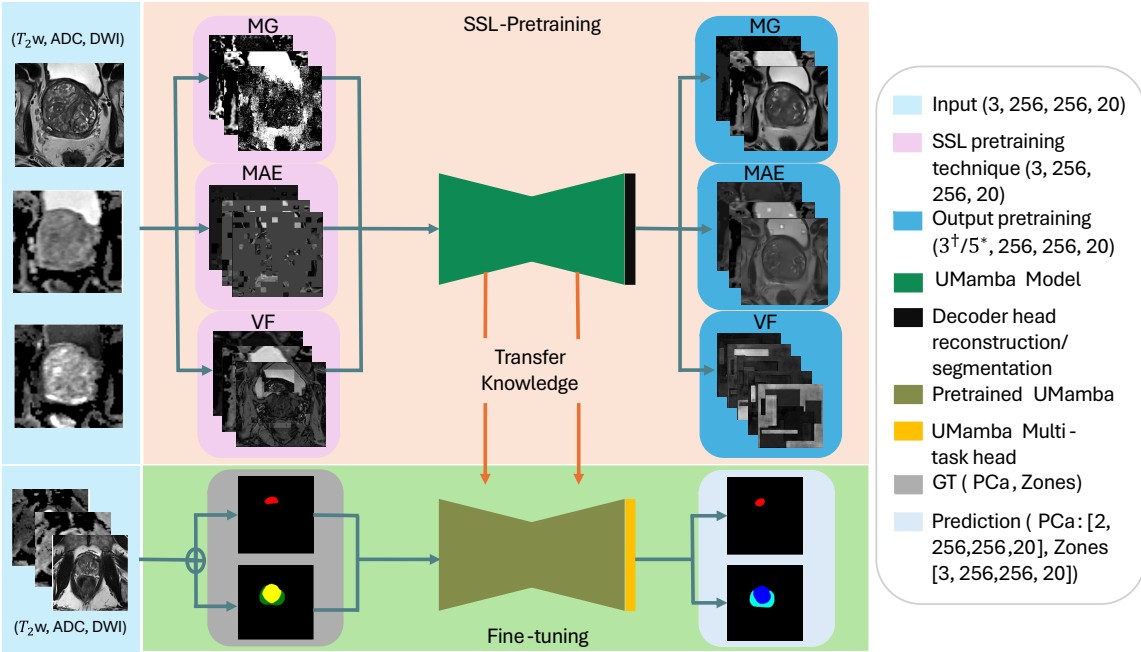

Figure 1: The UMamba model is pretrained separately using three SSL objectives; Model Genesis (MG), Masked Autoencoders (MAE), and Volume Fusion (VF). The output channel configurations are $3^{\dagger}$ for MG and MAE, and $5^{*}$ for VF. The pretrained UMamba model is subsequently fine-tuned using a multi-task objective that jointly predicts prostate cancer and prostate zones (peripheral and transitional zones).

### 2.1. Dataset

In this study, we utilized a total of 5,189 prostate MRI scans for self-supervised pretraining, fine-tuning, and evaluation. The dataset includes scans (N=2,431) derived from an institutional cohort at St. Olavs hospital (STOH), the PI-CAI public training and development set (1,500 cases), the PI-CAI hidden tuning cohort (100 cases), the PI-CAI hidden testing cohort (1,000 cases), and the external OOD Prostate158 (N=158) dataset. Across all datasets, positive cases are defined as histologically-confirmed clinically significant (ISUP $\geq$ 2), and negatives are determined based on histology (ISUP $\leq$ 1) or MRI (PI-RADS $\leq$ 2) findings, with a follow-up period of at least three years, except for in-house unlabeled data and P158. An overview of the datasets used across all stages is illustrated in Table 1 and the clinical variables associated with the in-house unlabeled cohort are provided in Appendix A.

For a detailed description of the inclusion and exclusion criteria, annotation methods, and clinical variables within the PI-CAI dataset, please refer to (Saha et al., 2024). Further descriptions of the in-house dataset and a short summary of PI-CAI are provided below.

The **In-House** dataset comprises prostate MRI exams from St. Olavs Hospital, Trondheim University Hospital (STOH)/ NTNU, Norway, collected between June 2014 and July 2024. Inclusion was based on suspicion of PCa via elevated PSA, digital rectal exam, repeated biopsy, or active surveillance. For patients with multiple scans, only the first available diagnostic scan at initial suspicion or diagnosis was retained. Scans obtained for in-bore biopsy, staging, or post-treatment follow-up were excluded, as well as scans with missing sequences or extraction errors. Of these, 2,431 unlabeled scans were used for self-supervised pretraining. The Regional Committee for Medical and Health Ethics, Mid-Norway, approved the use of the in-house dataset (identifier 2017/576). An additional dataset of 200 cases with expert annotations (Krüger-Stokke et al., 2021) of csPCa and prostate zones was excluded because they were partly (197/200) included in the PI-CAI test set. These cases were annotated using ITK-SNAP for csPCa and zonal anatomy by a radiology resident with two years of experience, supervised by a senior radiologist with over 10 years of experience. Moreover, the expert zonal segmentations were also used to evaluate the predictions of the prostate zones. Figure 4 in Appendix A further illustrates the selection process.

The **Public Training and Development Set** is a subset of the PI-CAI challenge and consists of 1,500 cases, of which 425 are csPCa. Of these cases, 220 were annotated by human experts, while the remaining annotations were generated using validated AI-based pipelines provided by the PI-CAI challenge organizers, as described in (Bosma et al., 2021). Additionally, prostate zonal annotations were provided, generated by training nnUNet (Isensee et al., 2021) on the ProstateX dataset (Yuan et al., 2025), a subset of the PI-CAI training set.

The **PI-CAI Hidden Tuning Set** consists of 100 cases, 46 of which are clinically significant PCa. We used this cohort to evaluate all trained models for performance comparison and to facilitate the selection of the best model.

The **PI-CAI Hidden Test Set** is a large-scale testing cohort comprising 1,000 cases, of which 398 are csPCa. These cases originate from eight sites across four centers in the Netherlands and one center in Norway. The best performing model, selected from the PI-

| Dataset | Cases | Type | Annotations | Stages |
|---|---|---|---|---|
| In-House | 2,431 | 3T bpMRI | None | SSL Pretraining |
| PI-CAI Training and Development set | 1,500 | 1.5T, 3T bpMRI | PCa[1], Zonal[2] | Fine-tuning, Ablation |
| PI-CAI Hidden tuning cohort | 100 | 1.5T, 3T bpMRI | PCa | Testing |
| PI-CAI Hidden testing cohort | 1,000 | 1.5T, 3T bpMRI | PCa | Testing |
| Prostate158 | 158 | 3T | PCa | Testing |

Table 1: Dataset information and usage across stages of the framework. [1]Includes 200 AI-generated PCa labels; [2]All zonal labels are AI-generated.

CAI hidden tuning set, was evaluated on this cohort as well as its non-SSL counterpart to assess the effect of pretraining.

The **Prostate158 (P158) dataset** (Adams et al., 2022) is a publicly available, expert-annotated external cohort comprising 158 biparametric 3T prostate MRI scans acquired using Siemens scanners. Each case includes $T_2$-weighted and ADC sequences with voxel-wise annotations of zonal anatomy and csPCa lesion defined as PI-RADS $\geq 4$ and ISUP$\geq 1$, performed in ITK-SNAP by two radiologists (6 and 8 years of experience). Notably, P158 differs from the PI-CAI hidden test set in terms of how the clinical significance of PCa is defined and is therefore used only to assess model generalizability on an external OOD cohort, rather than as a benchmark dataset.

## 2.2. Architecture

UMamba (Ma et al., 2024), an efficient adaptation of the Mamba framework (Gu and Dao, 2024), models long-range dependencies with linear time complexity as an alternative to convolutional and transformer-based networks. Our prior work (Larsen et al., 2025) introduced UMamba-MTL, extending the UMambabot variant with a multi-task single-decoder framework that integrates anatomical zone segmentation of the peripheral zone (PZ) and transition zone (TZ) as an auxiliary task alongside clinically significant prostate cancer (csPCa) detection. UMamba-MTL achieved state-of-the-art csPCa detection on an out-of-distribution in-house dataset (N=200), surpassing CNN and hybrid CNN-transformer models, and showed promising results on the PI-CAI Hidden tuning cohort (N=100). Motivated by these outcomes, we use UMamba for pretraining and UMamba-MTL for finetuning in this study; details are available in (Larsen et al., 2025).

## 2.3. Pretraining

Consider $D_u = \{X_u\}$ to be a large set of unlabeled 3D MRI volumes, where each volume $X \in \mathbb{R}^{C \times D \times H \times W}$, with $C$ denoting the number of imaging channels (T$_2$w, ADC, HBV), and

$D \times H \times W$ representing the volumetric spatial dimensions. We randomly initialize a 3D UMamba model (Ma et al., 2024), comprising an encoder $f_\theta$ and a task-specific head $f_\varphi$, and train it on $D_u$ by optimizing one of several self-supervised objectives $\mathcal{L}_{\text{SSL}}$, depending on the chosen pretext task. The aim is to encode meaningful anatomical and semantic features from the multi-modal MRI volumes by solving a proxy task. Formally, self-supervised pretraining aims to learn an encoder function $f_\theta : X \mapsto z$, where $X \sim D_u$ is an unlabeled input volume, and $z$ is its latent representation. A task-specific decoder $f_\varphi$ maps this representation to an output, which is trained to approximate a predefined target $T(X)$ derived from $X$.

$$\min_{\theta,\varphi} \mathbb{E}_{X \sim D_u} \left[ \mathcal{L}_{\text{SSL}} \left( f_\varphi(f_\theta(X)), T(X) \right) \right]$$

Here, $T(X)$ is the task-specific target: for reconstruction-based objectives , $T(X) = X$; and for pseudo segmentation-based tasks (Volume Fusion), $T(X)$ is a voxel-wise label map derived from the fusion process.

**Volume Fusion**, as presented in (Wang et al., 2023), is based on a pseudo-segmentation pretext task, where two sub-volumes are fused using different fusion categories. The model takes the fused volume as input and predicts the fusion category of each voxel. This pretraining strategy encourages the model to learn fine-grained spatial and semantic details.

**Model Genesis** is a unified self-supervised framework proposed in (Zhou et al., 2021), which corrupts 3D medical images using four types of augmentations. The model is then trained to reconstruct anatomical patterns from these distorted inputs. By learning such representations, the model becomes more generalizable across different organs, diseases, and imaging modalities.

**Masked Autoendoers** are a self-supervised method for learning representations by masking parts of the input image and reconstructing them in an autoencoder fashion (He et al., 2022). The reconstruction of occluded regions encourages the model to learn meaningful image representations and anatomical context. We train the 3D UMamba model (Ma et al., 2024) using an $L_2$ loss computed over the masked regions with a masking ratio of 75%, inspired by the work of (He et al., 2022). Further methodological details and formulations for each pretraining strategy are provided in Appendix B.

## 2.4. Fine-Tuning

The pretrained models are fine-tuned using the multi-task UMamba framework, following the setup in (Larsen et al., 2025). For reconstruction-based pretraining (MG, MAE), we replace the pretrained decoder's final convolutional layers with a segmentation head. Each Conv3D layer in this head is reinitialized to output $C_{\text{out}} = 5$ channels for PCa and zone prediction, whereas Volume Fusion–pretrained models already produce compatible multi-channel segmentation outputs and therefore require no adaptation. For fine-tuning, we apply the composite multi-task loss formulation introduced in (Larsen et al., 2025):

$$\mathcal{L}_{\text{csPCa}} = \mathcal{L}_{\text{Focal}}, \quad \mathcal{L}_{\text{Zonal}} = \lambda \mathcal{L}_{\text{Dice}} + (1 - \lambda)\mathcal{L}_{\text{CE}}, \quad \mathcal{L}_{\text{Total}} = \mathcal{L}_{\text{csPCa}} + \beta \mathcal{L}_{\text{Zonal}},$$

where $\lambda = 0.5$ balances Dice and Cross-Entropy losses, and $\beta = 0.2$ adjusts the relative weight of the zonal segmentation task. This formulation ensures effective fine-tuning across both the highly imbalanced csPCa detection task and the anatomical zonal delineation objective.

## 2.5. Comparison with other methods

We compare our approach against three baselines: transformer based Swin-UNETR model, a plain UMamba model, and Spark3D, a SOTA CNN-based SSL method derived from the nnU-Netv2 architecture (Wald et al., 2025) and modified for MAE pretraining. For plain UMamba, three pretrained models (VF, MG, MAE) were fine-tuned using Focal Loss (Lin et al., 2017).

The Spark3D and Swin-UNETR baselines were adapted to three-channel bpMRI input and pretrained using an MAE-based reconstruction objective. Spark3D utilizes a residual encoder U-Netv2 backbone (Isensee et al., 2024), while Swin-UNETR employs a hybrid CNN–Transformer architecture featuring hierarchical Swin Transformer encoding and convolutional decoding (Hatamizadeh et al., 2021). Both models were fine-tuned using Focal Loss (Lin et al., 2017), with weights for both the encoder and decoder transferred from the pretrained state. To ensure a fair comparison, model checkpoints for both baselines were selected using the same evaluation metric and methodology as our proposed method.

## 2.6. Evaluation Metric and Implementation details

**Evaluation Metric** Following the PI-CAI challenge guidelines (Saha et al., 2024), we evaluated model performance using the PI-CAI score, which is defined as the mean of the Average Precision (AP) reflecting lesion-level detection performance and the Area Under the Receiver Operating Characteristic Curve (AUROC) indicating patient-level diagnosis. The metrics are computed based on extracted non-overlapping lesion detection maps (Saha et al., 2024; Bosma et al., 2023), using the softmax output of the models. The PI-CAI score is given by

$$\text{score} = \frac{\text{AP} + \text{AUC}}{2}$$

In addition, we report a clinically relevant operating point using Free-Response ROC (FROC) analysis for the PI-CAI hidden test set (N=1000), specifically sensitivity (Sens3), which measures lesion-level sensitivity at a radiologist-equivalent false-positive rate (PI-RADS $\geq 3$). This provides a clinically interpretable measure linked to planning of targeted biopsies.

**Implementation details** Diffusion weighted images (DWI: ADC, HBV) were resampled to $T_2$w resolution before pretraining and fine-tuning. A patch size of $256 \times 256 \times 20$ was extracted, centered on the prostate using cropping, with zero padding or reflect padding applied as needed. For pretraining and fine-tuning we utilized PyTorch (Paszke et al., 2019), MONAI (Cardoso et al., 2022), nnU-Netv2 (Isensee et al., 2024) and nnSSL (Wald et al., 2025) frameworks.

For pretraining, the in-house dataset was split into 95% training (N=2,331) and 5% validation (N=100). The model received three channel inputs with augmentations tailored to each pretext task, following protocols described in (Wald et al., 2025). Pretraining was performed on an NVIDIA A40 GPU for 700 epochs with z-score normalized input using SGD with a polynomial learning rate scheduler. Spark3D and Swin-UNETR were trained with an MAE objective, incorporating architectural and patch-size modifications, respectively, to address bpMRI anisotropy (see Appendices D and E for details).

For fine-tuning, models were trained on the PI-CAI public training set (N=1,500), split into 80% training and 20% validation using five-fold cross-validation. Five-fold cross-validation was performed using a mean ensemble of softmax outputs, and prostate detection maps were subsequently generated using the lesion candidate extraction method described in (Saha et al., 2024; Bosma et al., 2023). The PCa detection maps were obtained using the same procedure across all comparative methods. All fine-tuned five-fold cross-validated models were wrapped in Docker containers for submission to the PI-CAI challenge forum.

Both UMamba and UMamba-MTL backbones were trained for 130 epochs, as models typically converged by 100 epochs, following (Larsen et al., 2025). Weights from both the encoder and decoder were transferred, with a warm-up phase of up to 10 epochs, during which the encoder was gradually unfrozen. Fine-tuning used the AdamW optimizer, a learning rate $1 \times 10^{-4}$, a batch size of 8, a cosine annealing scheduler, and random augmentations (Larsen et al., 2025). An ablation study of fine-tuning strategies is detailed in section 3.2.2

Similarly, MAE pretrained Swin-UNETR was fine-tuned following the same protocol as the UMamba models, while its randomly initialized counterpart was trained from scratch using the same training setup; for both models, the patch size was set to $256 \times 256 \times 32$ to satisfy architectural requirements.

Spark3D fine-tuning on the PI-CAI training set followed the same training hyperparameters reported in (Wald et al., 2025) using a five-fold split and training for 1000 epochs, with a warmup phase of 12.5k iterations ($\sim$50 nnU-Net epochs). Training employed SGD with momentum 0.99, a learning rate $1 \times 10^{-3}$, a batch size of 3, a polynomial scheduler, and a weight decay of $3 \times 10^{-5}$. The Spark3D (ResEnc-UNet/nnU-Netv2) without SSL was trained similarly but with a higher learning rate $1 \times 10^{-2}$. All fine-tuning was conducted on a single NVIDIA A100 GPU using the IDUN cluster (Själander et al., 2019).

## 3. Results

### 3.1. Quantitative and Qualitative results

We trained multiple self-supervised learning (SSL) objectives across all backbones (nnU-Netv2,Swin-UNETR, UMamba, and UMamba-MTL), followed by fine-tuning, as detailed in section 2.3 and section 2.4. Based on the five-fold cross-validated results on the PI-CAI hidden tuning cohort (N=100), UMamba-MTL with MAE pretraining (**UMamba-ProSSL**) achieved the highest PI-CAI score, as shown in Table 2. This model was subsequently evaluated on the PI-CAI hidden testing cohort (N=1,000). For comparison, a non-SSL UMamba-MTL model was also tested, and the results are reported in Table 3. Additionally, all SSL and non-SSL models across all backbone architectures were evaluated on the external OOD Prostate158 (P158) dataset, and the corresponding results are presented in Table 4.

Overall results indicate that, among SSL strategies, all methods improved downstream PI-CAI performance except for volume fusion, with MAE providing the most consistent and substantial gains across pretext tasks. Moreover, in terms of backbone performance, UMamba-MTL consistently outperformed UMamba, nnU-Netv2 and Swin-UNETR.

To qualitatively illustrate predictions, Figure 2 shows csPCa detection maps generated by UMamba-ProSSL on in-house $T_2w$, ADC, and HBV images, along with the corresponding prostate and zonal mask predictions.

Table 2: Performance of nnU-Netv2, SWin-UNETR UMamba, and UMamba-MTL with different SSL pretraining strategies on the PI-CAI hidden tuning cohort. PI-CAI Score, AUC, and AP are reported with 95% confidence intervals, along with the corresponding leaderboard ranks. † denotes the best-performing model, referred to as **UMamba-ProSSL**.

| Model | SSL Technique | PI-CAI Score | AUC | AP | Rank |
|---|---|---|---|---|---|
| nnU-Netv2 | Scratch | 0.710 (0.603–0.818) | 0.820 (0.725–0.903) | 0.601 (0.452–0.756) | 221$^{st}$ |
| | **Spark3D** | **0.736** (0.631–0.834) | **0.841** (0.754–0.913) | **0.631** (0.484–0.771) | **155$^{st}$** |
| Swin-UNETR | Scratch | 0.665 (0.556–0.772) | 0.792 (0.692–0.883) | 0.537 (0.393–0.682) | 269$^{st}$ |
| | **MAE** | **0.699** (0.593–0.798) | **0.805** (0.714–0.855) | **0.594** (0.453–0.729) | **235$^{st}$** |
| UMamba | Scratch | 0.735 (0.631–0.826) | 0.843 (0.760–0.914) | 0.627 (0.483–0.751) | 156$^{th}$ |
| | Volume Fusion | 0.716 (0.611–0.818) | 0.827 (0.739–0.902) | 0.605 (0.457–0.756) | 205$^{th}$ |
| | Model Genesis | 0.738 (0.634–0.835) | 0.835 (0.749–0.911) | 0.641 (0.499–0.733) | 148$^{th}$ |
| | **MAE** | **0.773** (0.671–0.866) | **0.862** (0.777–0.933) | **0.685** (0.546–0.813) | **42$^{nd}$** |
| **UMamba -MTL (ours)** | Scratch | 0.781 (0.689–0.865) | 0.867 (0.791–0.931) | 0.696 (0.564–0.813) | 34$^{th}$ |
| | Volume Fusion | 0.750 (0.652–0.839) | 0.868 (0.794–0.931) | 0.631 (0.493–0.760) | 93$^{rd}$ |
| | Model Genesis | 0.794 (0.703–0.875) | 0.888 (0.818–0.944) | 0.701 (0.573–0.816) | 22$^{nd}$ |
| | **MAE†** | **0.818** (0.730–0.898) | **0.914** (0.852–0.963) | **0.722** (0.592–0.846) | **1$^{st}$** |

The results for auxiliary prostate and zonal mask predictions on the in-house dataset (N = 200) are also reported in terms of the dice similarity coefficient (DSC) and are provided in Appendix C.

### 3.2. Ablation studies

We conducted two key ablation studies using the PI-CAI public training set, evaluating performance using the mean results across five cross-validation folds. First, we assessed the benefits of large-scale pretraining by artificially reducing the proportion of labeled data. Second, we investigated fine-tuning strategies to determine the optimal approach for transferring learned weights to the downstream task, evaluating different combinations of encoder freezing, decoder initialization, and learning rate selection.

Table 3: Results for UMamba-ProSSL and UMamba-MTL on the PI-CAI open development testing set (N=1,000). PI-CAI score, AUC, and AP are reported with 95% confidence intervals along with corresponding leaderboard ranks. Sensitivity (Sens3) corresponds to lesion-level sensitivity at the radiologist-equivalent operating point (PI-RADS $\geq$ 3) derived from the FROC analysis.

| Model | SSL Technique | PI-CAI Score | AUC | AP | Sens3 | Rank |
|---|---|---|---|---|---|---|
| UMamba-ProSSL | **MAE** | **0.780** (0.747–0.813) | **0.905** (0.885–0.924) | **0.655** (0.603–0.706) | **0.761** | **1st** |
| UMamba-MTL | Scratch | 0.776 (0.704–0.807) | 0.896 (0.875–0.916) | 0.656 (0.606–0.704) | 0.736 | 2nd |

Table 4: PI-CAI socre performance of nnU-Netv2, Swin-UNETR, UMamba, and UMamba-MTL with different SSL pretraining strategies on the external out-of-distribution P158 dataset.

| Model | SSL Technique | PI-CAI Score | AUC | AP |
|---|---|---|---|---|
| nnU-Netv2 | Scratch | 0.653 | 0.831 | 0.476 |
| | **Spark3D** | **0.701** | **0.850** | **0.552** |
| Swin-UNETR | Scratch | 0.633 | 0.803 | 0.463 |
| | **MAE** | **0.639** | **0.782** | **0.496** |
| UMamba | Scratch | 0.705 | 0.851 | 0.559 |
| | Volume Fusion | 0.675 | 0.793 | 0.557 |
| | Model Genesis | 0.704 | 0.853 | 0.556 |
| | **MAE** | **0.715** | **0.832** | **0.597** |
| **UMamba-MTL** | Scratch | 0.716 | 0.828 | 0.603 |
| | Volume Fusion | 0.697 | 0.829 | 0.566 |
| | Model Genesis | 0.721 | 0.836 | 0.607 |
| | **MAE** | **0.746** | **0.845** | **0.647** |

### 3.2.1. Effect of large-scale pretraining

In many medical domain applications, particularly in csPCa detection, labeled data is often scarce. Even within the PI-CAI public training and development sets, human annotation data is available for only 220 cases. SSL-based pretraining can ease the burden on radiologists and achieve higher performance than models trained from scratch using the same labeled data. To measure the effect of large-scale pretraining, we artificially reduced the number of images in the PI-CAI training and development set by percentage (10%, 30%, 50%, and 70%) to simulate low-data regimes. This reduction was achieved by stratifying the data based on ISUP grades and human annotations. Specifically, cases were first grouped by ISUP grade, and for clinically significant cases (ISUP ¿ 1), an additional stratification was applied based on whether expert lesion annotations were available. Stratified sampling was then performed to ensure that each reduced subset preserved the original distribution of

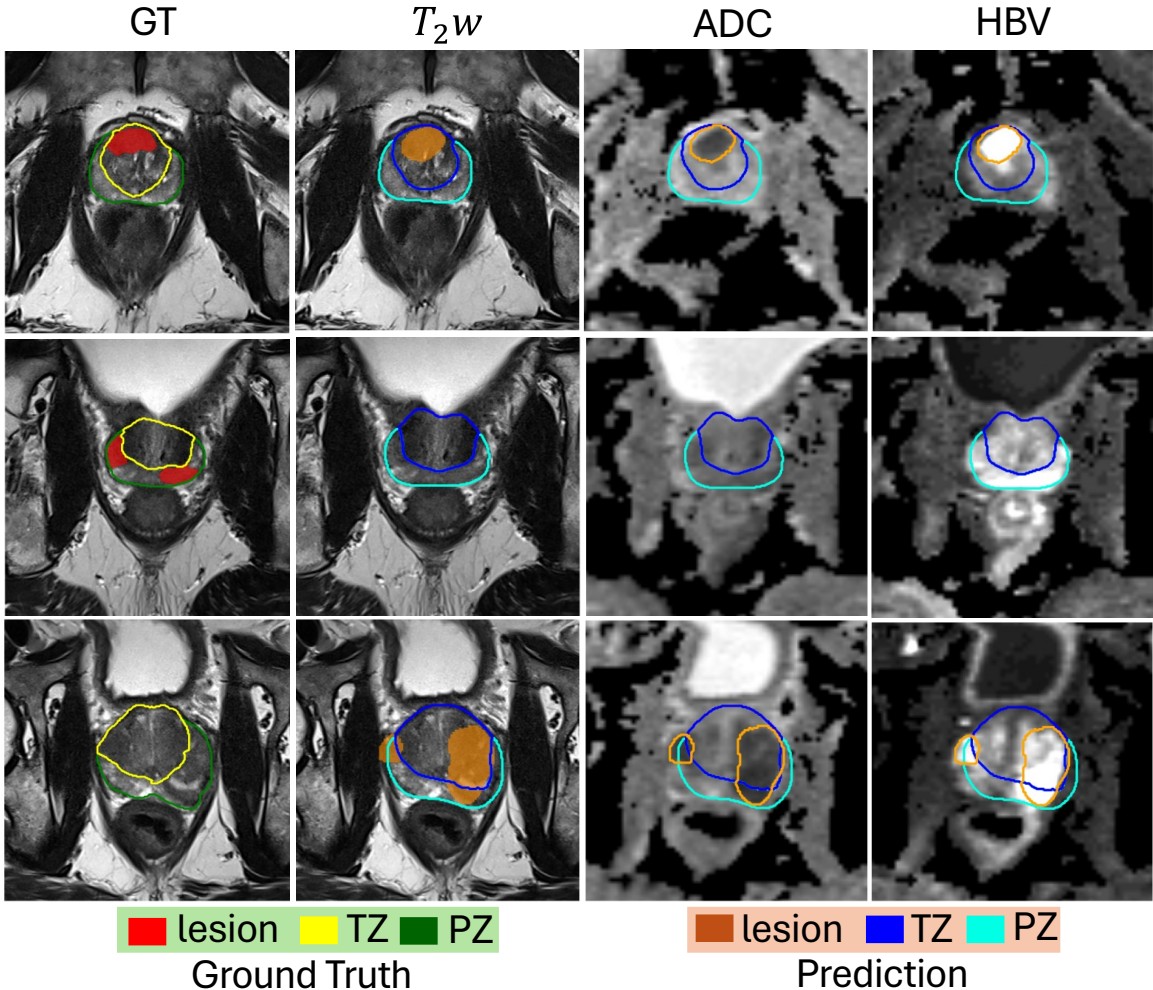

Figure 2: Qualitative results for UMamba-ProSSL on the in-house dataset (N=200). Ground truth (GT) annotations for clinically significant prostate cancer (csPCa) and the prostate zones (peripheral zone (PZ) and transition zone (TZ)) are overlaid on $T_2w$ images. Model predictions for csPCa and zonal anatomy are overlaid on $T_2w$, ADC, and HBV modalities. The orange contours on the ADC and HBV images illustrate the lesion hypointensities and hyperintensities, respectively. The first row shows a true positive (TP) case, the second row shows a false negative (FN) case, and the last row shows a false positive (FP) case.

disease severity and the proportion of human-annotated cases. The outcomes are presented in Figure 3.

The pretrained model achieved a substantially higher score than its counterpart trained from scratch when utilizing only 120 images. Notably, the pretrained model attained a

superior score with just 50% of the data ($N = 600$) compared to the scratch model using 70% of the data ($N = 840$). Ultimately, with the utilization of the full 100% dataset, the scratch model finally overcame its initialization by leveraging the strong supervisory signals provided by the large data size. While the scratch model's final score is extremely close, the pretrained approach exhibits superior convergence and robust generalization, as substantiated by the findings in Table 2 and Table 3.

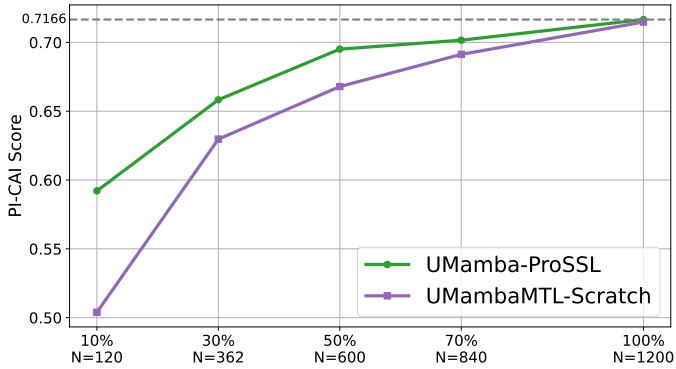

Figure 3: Comparison of PI-CAI scores between UMamba-ProSSL and UMambaMTL-Scratch, shown across varying labeled set sizes. Each point represents the mean value over five cross-validated folds.

### 3.2.2. Fine-tuning strategies

We investigated different fine-tuning strategies on our best-performing model, UMamba-ProSSL, to determine the optimal approach for transferring learned weights to the downstream task. We evaluated these strategies by varying decoder initialization and encoder freezing during warmup, resulting in four ablation settings. The number of warmup epochs was empirically set to 10, determined by translating the warmup iterations from (Wald et al., 2025) into relative epochs for our setup. We also explored different maximum learning rates and results are summarized in Table 5.

The results show a significant performance drop when the decoder is randomly initialized and the encoder is frozen during the warmup period. Furthermore, transferring both encoder and decoder weights while freezing the encoder early also leads to a performance decline. Lastly, using a smaller learning rate during fine-tuning does not necessarily result in improved performance.

## 4. Discussion and conclusion

In this study, we show that the UMamba-MTL backbone with MAE pretraining (UMamba-ProSSL) achieves state-of-the-art PCa detection on bpMRI. It demonstrates superior performance on the PI-CAI hidden tuning cohort (N=100), the large-scale, multi-center, multi-site hidden testing cohort (N=1,000), and OOD external P158 (N=158) datasets . Among these,

Table 5: Comparison of different fine-tuning strategies for the UMamba-ProSSL on the PI-CAI public training and development set. Results assess the impact of Encoder/Decoder pretraining and freezing alongside variations in the Maximum Learning Rate (Max. LR). Performance is reported as the mean PI-CAI score across five cross-validation folds.

| Encoder Pretrain | Encoder Freeze during warmup | Decoder Pretrain | Max. LR | PI-CAI Score |
|---|---|---|---|---|
| Yes | No | No | $1e^{-4}$ | 0.710 |
| Yes | Yes | No | $1e^{-4}$ | 0.639 |
| Yes | Yes | Yes | $1e^{-4}$ | 0.682 |
| Yes | No | Yes | $1e^{-3}$ | 0.715 |
| **Yes** | **No** | **Yes** | **$1e^{-4}$** | **0.716** |
| Yes | No | Yes | $1e^{-5}$ | 0.712 |

the PI-CAI leaderboard is the most comprehensive benchmark for PCa detection available to date.

On the PI-CAI hidden tuning set[2], UMamba-ProSSL obtained a PI-CAI score of 0.818 (95% CI: 0.730–0.898), an AUC of 0.914 (95% CI: 0.852–0.963), and an AP of 0.722 (95% CI: 0.592–0.846), achieving a 3.02% relative gain in the PI-CAI score over model genesis (Zhou et al., 2021) and a 0.5% gain over the second-ranked method on the leaderboard. At the time of evaluation, these results placed UMamba-ProSSL 1st among 775 entries, highlighting its competitiveness. The synergy between MAE pretraining and the UMamba backbone leads to tangible performance improvements. Notably, MAE pretraining also boosts the plain UMamba model's performance, resulting in a 2.50% relative gain in AUC and a substantial 8.56% relative gain in AP compared to training from scratch. While MAE pretraining enhances the Spark3D model (Wald et al., 2025) and Swin-UNETR (Hatamizadeh et al., 2021) performance relative to their baseline, a gap in PI-CAI score persists compared with our proposed model. This disparity may be due to the anisotropic nature of prostate imaging and thus further supports our choice of using a strong backbone architecture for robust performance. We also observed that volume fusion (Wang et al., 2023), a segmentation-based pretraining optimizing dice metric, is not optimal for the downstream PCa detection task, especially considering the ineffectiveness of the dice score for evaluating multifocal PCa lesions (Yan et al., 2022).

UMamba-ProSSL was further assessed on the PI-CAI hidden test set[3] (N=1,000), attaining a PI-CAI score of 0.780 (95% CI: 0.747–0.813), an AUC of 0.905 (95% CI: 0.885–0.924), and an AP of 0.655 (95% CI: 0.603–0.706). Also, at a clinically relevant operating point corresponding to a radiologist-equivalent false-positive per examination (PI-RADS $\geq$ 3), the model achieved a lesion-level sensitivity (Sens3) of 0.761, the highest among all leaderboard entries, compared with a sensitivity of 0.961 reported for human readers on the PI-CAI hidden test set. These results positioned UMamba-ProSSL highest among the 45 submis-

---

2. https://pi-cai.grand-challenge.org/evaluation/open-development-phase/leaderboard/

3. https://pi-cai.grand-challenge.org/evaluation/challenge/leaderboard/

sions on the leaderboard, surpassing all previous CNN, nndetection, and transformer-based methods. Critically, this result was achieved on a rigorously designed multi-center cohort from eight sites using Siemens and Philips MRI scanners, featuring significant heterogeneity in patient demographics, disease prevalence, and imaging acquisition protocols. Securing first place under these real-world retrospective conditions demonstrates UMamba-ProSSL's robustness, strong clinical generalizability and sensitivity in this European setting.

On the PI-CAI hidden test set, the baseline UMamba-MTL model also performed commendably, achieving a PI-CAI score of 0.776 (95% CI: 0.704-0.807), an AUC of 0.896 (95% CI: 0.875-0.916), and an AP of 0.656 (95% CI: 0.606-0.704). These results further signify the nature of the UMamba architecture in capturing long-range dependencies through the integration of zones in the auxiliary task. Although the absolute performance difference between UMamba-ProSSL and the scratch UMamba-MTL model on the hidden test set is modest, overall gain by pretraining is consistent across multiple complementary metrics and evaluation settings. Notably, UMamba-ProSSL achieves higher sensitivity at the clinically relevant operating point (Sens3) and more than 3% performance gain over its non-pretrained counterpart on the OOD external P158 dataset, further highlighting the model's clinical relevance, generalizability and robustness.

The qualitative results presented in Figure 2 show that csPCa detection is a challenging task. The false negative cases are often attributed to isointensities in the ADC and DWI modalities, which obscure true cancerous lesions. Conversely, false positives might reflect prostatitis or other benign findings that exhibit hypo- and hyperintensities in the ADC and high b-value (HBV) DWI modalities, respectively, leading the model to incorrectly identify them as true lesions. For true positive lesions, our method (UMamba-ProSSL) achieved a high average precision (AP), demonstrating high lesion-level precision across both the PI-CAI hidden tuning and hidden testing sets. The high precision is crucial for guiding biopsy procedures, as the patient-level AUC does not reflect lesion localization. High precision directly improves the urologist's ability to sample correctly during biopsy, thereby enhancing diagnostic accuracy and reducing unnecessary procedures.

The effectiveness of the large-scale pretraining approach, particularly its observed label efficiency in the low-data regime, directly addresses clinical settings where vast archives of routine, unlabeled bpMRI images are available. Our SSL method using MAE leverages this data to learn generalizable representations before fine-tuning, thereby improving csPCa detection while reducing reliance on scarce and expensive expert annotations.

Several limitations of our study guide future investigations. First, regarding comparative performance, although the nnU-Netv2 model by (Pooch et al., 2025) was trained with substantially more human labels (425 human vs. our 220 human and 205 AI annotations), it achieved lower performance on the same benchmark. Future work should include an analysis using fully human-labeled data to further isolate the effect of large-scale pretraining and better characterize performance in low-data regimes. Second, while our method cannot be directly compared to the reader study of (Saha et al., 2024) due to differences in cohort size and study design, valuable contextual insights can still be drawn. The study reported an AUC of 0.86 (95% CI: 0·83–0·89) on 400 cases, whereas our approach achieved an AUC of 0.905 (95% CI: 0·885–0·924) on 1,000 cases from the PI-CAI hidden test set. Although not directly comparable, this suggests that our model is competitive with expert readers

in patient-level diagnosis. Since our evaluation on the PI-CAI dataset is retrospective, prospective clinical studies are essential to fully assess the utility of our method.

Although we evaluated on the external Prostate158 (P158) dataset to assess OOD generalization, P158 does not follow the same reference standard as PI-CAI, particularly with respect to the explicit distinction between clinically significant and clinically insignificant PCa. To our knowledge, no other publicly available dataset currently matches the scale and standardized labeling of PI-CAI. Consequently, the lower absolute PI-CAI scores observed on P158 should be interpreted as a dataset-related limitation.

Finally, while MAE outperformed other SSL pretext tasks, future research should explore prostate-specific tasks that leverage anatomy and clinical data, while also investigating how performance scales with unlabeled data volume to identify potential diminishing returns. Furthermore, while we compared against hybrid CNN–Transformer architecture (Swin-UNETR) pretrained with an MAE objective, future work should explore pure transformer-based models and a broader range of self-supervised objectives, as the performance of Swin-UNETR was relatively limited in the context of prostate MRI.

In conclusion, UMamba-ProSSL achieves state-of-the-art performance on a large-scale benchmark by combining a UMamba-MTL backbone with MAE-based large-scale self-supervised pretraining, thereby advancing prostate cancer detection.

# Appendix A. Dataset

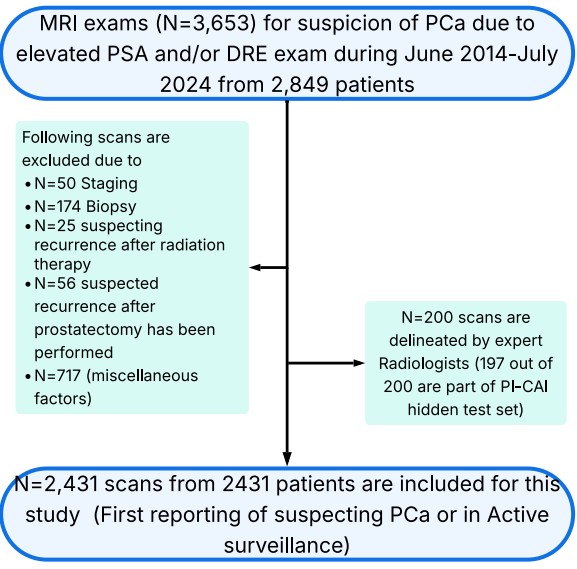

Figure 4: Inclusion and exclusion criteria for prostate MRI exams used in this study.

## A.1. Clinical variables

| Feature | Pretraining unlabeled data (STOH) |
| --- | --- |
| Sites | 1 |
| Patients | 2431 |
| Median age, years | 67 (61–72) |
| Median PSA, ng/mL | 7 (5–11) |
| Median prostate volume, mL | 47 (33–67) |
| Field strength, Tesla | 3 |
| Cases | 2431 |
| Positive MRI lesions | 1726 |
| PI-RADS 3 | 441 (25%) |
| PI-RADS 4 | 589 (34%) |
| PI-RADS 5 | 696 (40%) |
| Patient-level clinically significant PCa (GG $\geq$ 2) | 799 |

Table 6: Clinical variables for the St. Olavs Hospital (STOH) unlabeled cohort: Prostate Specific Antigen (PSA), Prostate Imaging-Reporting and Data System (PI-RADS), and Gleason Grade (GG).

## Appendix B. Pretraining

### B.1. Volume Fusion

Given two sub-volumes $I_b$ (background) and $I_f$ (foreground), generate a fused volume $X \in \mathbb{R}^{D \times H \times W}$ using a voxel-wise fusion map $\alpha \in \mathcal{V}$, as:

$$X = \alpha I_f + (1 - \alpha) I_b.$$

The corresponding label map $Y$ is derived from $\alpha$, and the model is trained to predict voxel-wise fusion classes using the following segmentation loss:

$$\mathcal{L}_{\text{sup}} = \frac{1}{2} (\mathcal{L}_{\text{dice}} + \mathcal{L}_{\text{ce}}).$$

For additional details, we refer the reader to the original reference.

### B.2. Model Genesis

The four augmentations include: (1) *Non-linear intensity transformations*, which monotonically distort voxel intensities to encourage the model to capture tissue appearance and contrast; (2) *Local pixel shuffling*, which permutes voxel positions within a local window, helping the model learn about textures and boundaries; (3) *Inner cutout* and (4) *Outer cutout*, which both involve masking parts of a sub-volume using arbitrarily shaped windows. In inner cutout, the central region is masked while the outer area is retained; in outer cutout, the opposite is done. These augmentations guide the model to interpolate or extrapolate missing information, promoting awareness of local and global anatomical continuity and geometry.

Consider a set of sub-volumes $\mathcal{X} = \{x_1, x_2, \ldots, x_n\}$ is extracted from raw 3D scans. These sub-volumes are transformed using a distortion function $f(\cdot)$, producing a set $\widetilde{\mathcal{X}} = f(\mathcal{X}) = \{\tilde{x}_1, \tilde{x}_2, \ldots, \tilde{x}_n\}$. The model is trained to reconstruct the original sub-volumes from the distorted ones. This reconstruction is formulated as learning a function $g(\cdot)$ such that:

$$g\left(\widetilde{\mathcal{X}}\right) = \mathcal{X} = f^{-1}\left(\widetilde{\mathcal{X}}\right).$$

The network minimizes the mean squared error (MSE) between the predicted output $\hat{x}_i = g(\tilde{x}_i)$ and the original input $x_i$:

$$\mathcal{L}_{\text{MG}} = \frac{1}{n} \sum_{i=1}^{n} \|\hat{x}_i - x_i\|_2^2.$$

### B.3. Masked Autoencoders

For a 3D U-Mamba model (Ma et al., 2024), let the raw input/target volume be $X \in \mathbb{R}^{B \times D \times H \times W \times C}$, the predicted output volume be $\hat{X} \in \mathbb{R}^{B \times D \times H \times W \times C}$, and a binary mask $M$ of the same shape, where $M = 1$ indicates visible (unmasked) voxels and $M = 0$ indicates masked voxels. The reconstruction loss based on the $L_2$ norm is defined as:

$$\mathcal{L}_{\text{MAE}} = \frac{\sum (1 - M) \cdot (\hat{X} - X)^2}{\sum (1 - M)}$$

Here, the numerator computes the squared reconstruction error only over masked voxels, and the denominator normalizes by the number of masked elements. This encourages the model to infer the missing regions solely from the visible context.

## Appendix C. Prostate and zonal segmentation results

One additional advantage of our model is that the auxiliary task also provides prostate segmentation. This can be clinically useful, as accurate segmentation of the prostate and its zones is important for biopsy guidance. In clinical practice, transrectal ultrasound (TRUS) guided biopsies are often performed using MRI–ultrasound fusion, in which $T_2$ weighted MRI is fused with ultrasound to enable MRI-targeted biopsies and improve lesion localization. Reported inter-reader variability of $DSC_{PZ} = 0.75$ and $DSC_{TZ} = 0.87$ was closely matched by our model, which achieved DSC scores of 0.76 for the peripheral zone (PZ) and 0.87 for the transition zone (TZ), respectively.

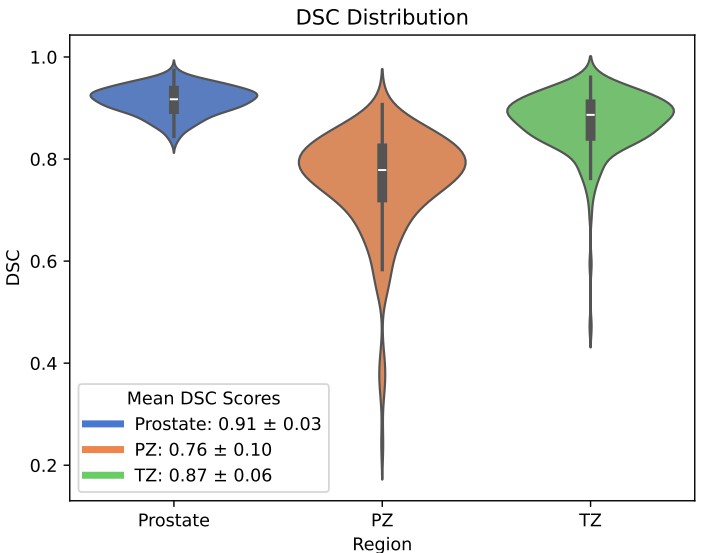

Figure 5: Prostate zonal segmentation (Peripheral Zone (PZ) and Transition Zone (TZ)) results on in-house St. Olavs Hospital cases (N=200).

## Appendix D. Details on Spark3D

| Hyperparameter | Value |
|---|---|
| features per stage | [32, 64, 128, 256, 320, 320] |
| norm op | torch.nn.InstanceNorm3d |
| nonlin | torch.nn.LeakyReLU |
| nblocks per stage | [1, 3, 4, 6, 6, 6] |
| conv op | torch.nn.Conv3d |
| nconv per stage decoder | [2, 2, 2, 2, 2] |
| kernel sizes | [[3,3,3], [3,3,3], [3,3,3], [3,3,3], [3,3,3], [3,3,3]] |
| nstages | 6 |
| strides | [[1,1,1], [1,2,2], [1,2,2], [2,2,2], [2,2,2], [1,2,2]] |
| network class name | ResidualEncoderUNet |

Table 7: Residual Encoder UNet (nnU-Netv2) configuration adapted for anisotropic bpMRI.

## Appendix E. Details on Swin-UNETR

| Hyperparameter | Value |
|---|---|
| network class name | SwinUNETR (MONAI) |
| in_channels | 3 |
| out_channels | 2 |
| img_size | $[256, 256, 32]$ |
| feature_size | 48 |
| use_v2 | True |
| depths (default) | $(2, 2, 2, 2)$ |
| num_heads (default) | $(3, 6, 12, 24)$ |
| patch_size (default) | $(2, 2, 2)$ |
| window_size (default) | $(7, 7, 7)$ |
| norm_name (default) | instance |
| spatial_dims (default) | 3 |

Table 8: Swin-UNETR configuration hyper-parameters used in this study.

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
