# OpenReview forum: "UMamba-ProSSL: Self-Supervised Large-Scale Pretraining with Multi-Task UMamba Advances Prostate Cancer Detection in Biparametric MRI"
_MIDL.io/2026/Validation_Papers — MIDL 2026 - Validation Papers Poster_

### Official Review · Reviewer_4jiw · 2025-12-22

**Confidence:** 4
**Preliminary Rating:** 5

**Summary:**

This paper presents UMamba-ProSSL, a large-scale self-supervised learning framework for clinically significant prostate cancer detection in biparametric MRI. The authors pretrain a strong UMamba backbone using multiple SSL objectives Model Genesis, Volume Fusion, and Masked Autoencoders on a sizable unlabeled in-house dataset, followed by multi-task fine-tuning for lesion detection and zonal segmentation. Extensive benchmarking on the PI-CAI hidden tuning and testing cohorts shows that MAE pretraining consistently delivers the best downstream performance, with the proposed method achieving first place on the PI-CAI leaderboard at the time of evaluation. The work demonstrates that pairing large-scale unlabeled prostate MRI with a modern backbone and MAE pretraining yields robust gains, particularly in low-label regimes.

**Strengths:**

The study is methodologically strong and clinically grounded. The use of a large, realistic unlabeled dataset for SSL pretraining directly addresses a key bottleneck in prostate MRI research. Evaluation on the PI-CAI hidden cohorts provides credible, unbiased evidence of generalization, and the leaderboard results substantiate the claimed state-of-the-art performance. The systematic comparison of multiple SSL pretext tasks under identical conditions is valuable and reinforces recent findings that MAE is particularly effective for 3D medical imaging. The integration of multi-task learning with zonal segmentation is well motivated clinically and contributes to improved lesion detection performance. Overall, the paper is thorough, well written, and reproducible, with code made publicly available.

**Weaknesses:**

Despite strong results, the methodological novelty is incremental, as the main contribution lies in combining established SSL objectives with an existing backbone rather than introducing a new learning paradigm. Comparisons are limited primarily to CNN-based baselines and Spark3D; transformer-based SSL methods are not explored, which narrows the scope of architectural insights. Some reported gains on the PI-CAI hidden test set over the non-SSL baseline are modest, raising questions about cost–benefit trade-offs given the substantial pretraining effort. Additionally, the reliance on AI-generated annotations in parts of the fine-tuning data may confound attribution of performance gains solely to SSL.

**Detailed Comments:**

The paper would benefit from a clearer discussion of computational cost versus performance gains, particularly given the relatively small absolute improvement over a strong scratch-trained UMamba-MTL model on the hidden test set. Clarifying whether MAE masking ratios or pretraining hyperparameters were tuned for prostate MRI specifically would improve transparency. A brief comparison or discussion of expected performance relative to transformer-based SSL approaches would further contextualize the contribution.

**Justification Of The Preliminary Rating:**

This paper makes a strong empirical contribution by demonstrating, at scale and on a highly competitive benchmark, that MAE-based self-supervised pretraining with a modern backbone yields state-of-the-art performance for prostate cancer detection. While methodological novelty is limited, the rigor of the evaluation, clinical relevance, and clear practical implications justify acceptance at MIDL.

**Questions To Address In The Rebuttal:**

How sensitive are the reported gains to the scale of unlabeled pretraining data, and is there evidence of diminishing returns beyond the current dataset size?
Do the authors expect MAE pretraining to provide similar advantages when paired with transformer-based backbones for prostate MRI?

---

### Official Review · Reviewer_HoxA · 2026-01-03

**Confidence:** 5
**Preliminary Rating:** 5
**Final Rating:** 5

**Summary:**

This research proposes UMamba-ProSSL, a self-supervised learning (SSL) framework for clinically significant prostate cancer (csPCa) detection in biparametric MRI (bpMRI), addressing labeled data scarcity and inter-observer variability in radiological diagnosis. Leveraging a large unlabeled dataset (N=2,431) and three SSL pretext tasks, it uses the UMamba-MTL backbone, with masked autoencoders (MAE) pretraining delivering the optimal performance. Evaluated on the PI-CAI benchmark, the framework secured first place on both hidden tuning and testing cohorts (N=1,000), achieving a well generalization across multi-center data. Its label efficiency and lesion localization highlight potential to reduce unnecessary biopsies and support clinical workflows.

**Strengths:**

The research leverages a large-scale unlabeled bpMRI dataset for SSL pretraining, addressing the scarcity of labeled medical data, and combines it with rigorous external validation on the multi-center PI-CAI benchmark, ensuring its relevance.
Its comparison of three state-of-the-art pretext tasks under identical training conditions provides insights into optimal SSL strategies for csPCa detection.

**Weaknesses:**

The in-house unlabeled dataset (N=2,431) lacks three-year follow-up confirmation, casting doubt on its consistent relevance to clinically significant prostate cancer and potentially undermining the reliability of SSL pretraining signals.
While acknowledging the domain-specificity of prostate anatomy, the authors did not develop prostate-specific pretext tasks.

**Detailed Comments:**

1. In Sec. 2.1, the authors note that the PI-CAI Training and Development set includes 200 AI-generated PCa labels and AI-derived zonal annotations, but fail to specify the validation metrics.
2. Sec. 2.3 describes Masked Autoencoder (MAE) pretraining but does not explicitly state the masking ratio used for 3D bpMRI volumes.
3. In Sec. 3.2.1’s label efficiency ablation, the authors mention stratifying the PI-CAI training data by ISUP grades and human annotations but do not detail the stratification methodology.

**Justification Of Final Rating:**

The work of this study aligns with the conference's theme of validation research, and the related work is highly solid. I am fully satisfied with the authors' responses, and thus I assign a final rating of 5.

**Justification Of The Preliminary Rating:**

This research fits MIDL 2026’s Validation Studies track. It solves prostate cancer detection’s labeled data scarcity via SSL pretraining, performs well on the multi-center PI-CAI benchmark, and shows the scientific rigor. Its multi-task design boosts clinical utility. Minor implementation clarifications are needed, but it meets the track’s core robustness requirements. I recommend a Strong Accept.

**Questions To Address In The Rebuttal:**

Please check the Weaknesses and Detailed Comments.

---

### Official Review · Reviewer_am8V · 2026-01-10

**Confidence:** 3
**Preliminary Rating:** 3
**Final Rating:** 5

**Summary:**

The paper proposes UMamba-ProSSL for csPCa lesion detection. It is a two-stage framework that first performs self-supervised pretraining on a UMamba backbone using a large unlabeled bpMRI cohort, and then fine-tunes a supervised multi-task model UMamba-MTL on PI-CAI labeled data for joint csPCa detection and prostate zonal prediction. The paper systematically benchmarks three SSL objectives and reports that MAE yields the best transfer to the downstream task. Overall, the MAE-pretrained UMamba-MTL achieves SOTA performance on the PI-CAI grand challenge benchmark.

**Strengths:**

1. Strong overall performance on the PI-CAI metric. The method achieves top-tier PI-CAI results, AP and AUC, which supports the claim that the model is not only classifying well but also localizing lesions reasonably well.
2. Detailed ablation studies. The paper includes ablations that are informative.
3. Training details are fairly thorough, which helps reproducibility.

**Weaknesses:**

1. In addition to nnU-Netv2, it would be helpful to compare against other commonly used 3D detection or segmentation baselines.
2. Clinical significance is still somewhat indirect. The paper reports strong benchmark performance on PI-CAI, AUC and AP, but it remains unclear how this translates to real-world clinical workflows.
3. Subgroup robustness remains unclear. The paper mentions multi-center hidden testing, but it does not provide a per-site (or per-scanner) performance breakdown, so it’s hard to assess how consistent the method is across centers.
4. No additional external validation beyond PI-CAI. The results are strong on the PI-CAI benchmark, but the paper does not include a separate, fully independent external cohort evaluation to further support real-world generalization.

**Detailed Comments:**

1. The comparison set feels a bit narrow for a validation-focused track. In addition to nnU-Netv2, consider adding some other baseline models, for example, standard 3D U-Net, or nnDetection. This would strengthen the claim that the observed gains are robust across baseline families.
2. The PI-CAI score, AUC and AP are useful, but clinical translation often depends on specific operating points. It would be helpful to report performance at clinically relevant thresholds. No prospective evaluation or reader study, so translation to workflow impact remains indirect.
3. To strengthen real-world generalization claims, it would be helpful to include one additional fully independent cohort with a clearly defined reference standard and report the same metrics.

**Justification Of Final Rating:**

The authors substantially strengthened the validation by expanding comparisons beyond nnU-Netv2, adding clinically interpretable operating-point reporting via FROC, and providing additional external evaluation on the P158 cohort. Together with the strong PI-CAI hidden-set performance and the thorough ablations, this offers a rigorous, reproducible, and practically relevant validation of SSL pretraining for csPCa lesion detection. While per-site breakdown on the hidden test set is not possible due to PI-CAI constraints, it does not materially detract from the paper’s main conclusions or its contribution as a robust benchmark-driven validation study.

**Justification Of The Preliminary Rating:**

The MAE-pretrained UMamba-MTL achieves top PI-CAI performance, suggesting the SSL choice is meaningful. And the ablation study is rigorous. However, key validation aspects remain unclear for this track: subgroup robustness is not reported, there is no additional independent external cohort beyond PI-CAI, and clinical significance is still indirect.

**Questions To Address In The Rebuttal:**

1. Can you add or justify at least one common baseline beyond nnU-Netv2?
2. In addition to reporting AUC/AP, can you provide clinically interpretable evaluation that better reflects real workflows? For example, a reader study showing whether the model improves radiologists’ lesion.
3. Do you have any additional external cohort for evaluation beyond PI-CAI?
4. Since PI-CAI is multi-center, can you provide a per-site breakdown of PI-CAI score, AP, and AUROC?

---

### Author Rebuttal · Authors · 2026-01-24

**Rebuttal:**

We thank the reviewers for their positive and thoughtful feedback. We appreciate the recognition of the systematic evaluation of SSL pretext tasks, particularly the effectiveness of MAE as well as the strong performance on the PI-CAI benchmark. We are also grateful for the acknowledgment of the study’s thorough experimentation, ablation analyses, reproducibility, and relevance to real-world prostate MRI analysis.

In response to the reviewers’ comments, we have addressed the identified weaknesses point by point and have attached the revised manuscript (with revisions highlighted in yellow). Major updates include the addition of a transformer-based baseline (Swin-UNETR) with MAE pretraining, out-of-distribution external evaluation, inclusion of a clinically motivated lesion-level sensitivity metric, clarification of previously missing implementation details, and an expanded discussion of methodological limitations and future research directions.

**Supporting Material:**

/attachment/35c3aa955d7f06795c5fd023fa1dc7c245526a89.pdf

---

### Meta-Review · Area_Chair_QYVG · 2026-02-05

**Recommendation:** Accept (Poster)
**Confidence:** 4

**Metareview:**

This paper presents UMamba-ProSSL, a self-supervised learning (SSL) framework for clinically significant prostate cancer detection in biparametric MRI. The work is of high quality, achieving SOTA performance on the authoritative PI-CAI benchmark, substantiated by hidden-cohort validation. It directly addresses the critical bottleneck of labeled data scarcity in medical imaging and demonstrates robust, label-efficient learning with clear clinical potential to reduce unnecessary biopsies. Noted the initial weaknesses raised by the reviewers include modest methodological novelty, limited comparisons to transformer-based SSL baselines, reliance on AI-generated annotations potentially confounding results, and an indirect clinical translation lacking subgroup robustness analysis and external validation beyond PI-CAI. Nonetheless, the reviewers were all satisfied with the clarification and additional results provided by the authors after the rebuttal. Eventually, reviewers agree, and AC also concur that the presented work is a clinically grounded and methodologically solid contribution that advances the application of foundation models in medical imaging. The additional results and promised revision are required in the final camera-ready.

---

### Decision · Program_Chairs · 2026-02-14

Accept (Poster)